# EBV Early Lytic Antigens, EBNA2 and PDL-1, in Progressive Multiple Sclerosis Brain: A Coordinated Contribution to Viral Immune Evasion

**DOI:** 10.3390/ijms27010437

**Published:** 2025-12-31

**Authors:** Lucia Benincasa, Barbara Rosicarelli, Chiara Meloni, Barbara Serafini

**Affiliations:** Department of Neuroscience, Istituto Superiore di Sanità, Viale Regina Elena 299, 00161 Rome, Italy; lucia.benincasa@guest.iss.it (L.B.); barbara.rosicarelli@iss.it (B.R.); chiara.meloni@guest.iss.it (C.M.)

**Keywords:** multiple sclerosis, Epstein-Barr virus, tertiary lymphoid organs, inhibitory immune checkpoint

## Abstract

Epstein-Barr virus (EBV) infection shows the strongest causative association with multiple sclerosis (MS), but its contribution to disease progression and the mechanisms allowing for viral persistence in the MS brain are still elusive. Studies in post-mortem MS brain tissue indicate an ongoing yet ineffective antiviral immune reaction in advanced stages of the disease. EBV has evolved strategies to evade immune recognition and clearance by the host immune system during both the latency and lytic phase of its life cycle. Recent evidence demonstrates that cells expressing EBV latent membrane protein (LMP) 2A exploit the PD-1/PDL1 inhibitory immune checkpoint to escape immune surveillance and maintain a persistent latent infection in the MS brain. This study investigated whether the virus also utilizes this inhibitory mechanism during other phases of the viral life cycle. By using multiple immunostainings on highly inflamed MS brain tissues containing meningeal tertiary lymphoid structures (TLSs), we analyzed PD-L1 expression on EBV-infected cells expressing EBNA2, five EBV lytic gene products, BZLF1, BHRF1, BMRF1, BALF2, and gp350/220, as well as on follicular dendritic cells within the TLSs. This is the first study describing in secondary progressive MS brain tissue the expression and the cellular and tissue distribution of PD-L1 on EBV-infected cells being in different stages of the viral life cycle, and confirms the meningeal TLSs as immune-permissive habitats favoring the maintenance of an intracerebral EBV reservoir.

## 1. Introduction

Multiple sclerosis (MS) is a chronic inflammatory/demyelinating disease of the central nervous system (CNS) causing progressive neurodegeneration.

The available disease modifying therapies (DMTs) effectively reduce relapses and delay disability worsening in the relapsing-remitting (RR) form of MS, but a substantial proportion of people with MS eventually convert to the secondary progressive (SP) phase of the disease for which DMTs provide limited or no benefit.

During the SP phase, the immune system activity shifts from peripheral inflammation to a compartmentalized response, largely confined within the CNS, mainly in the meninges, Virchow-Robin spaces, and areas around the chronic lesions, and sustains tissue damage despite reduced peripheral inflammatory activity.

Key features of the compartmentalized inflammation in MS brain include the formation of B and T cell aggregates resembling tertiary lymphoid structures (TLS) in the subarachnoid spaces [1,2], the presence of tissue resident memory (Trm) T cells [3,4], B cell-enriched infiltrates in the white matter (WM) lesions, and a persistent microglial/macrophage activation. Together, these elements sustain local immune activity and are associated with more severe disease progression [5].

TLSs are structures forming in non-lymphoid organs in response to persistent antigenic stimulation and chronic inflammation and represent sites of B cell expansion and immune activation [6]. TLSs have been observed in cerebral sulci of a substantial proportion (more than 40%) of SPMS donors [7]. Their characteristic can vary from B/T lymphocyte clusters to highly organized structures also containing plasma cells, follicular dendritic cells (FDCs), stromal cells, and germinal center-like areas. Unlike secondary lymphoid organs, intrameningeal TLS lack encapsulation and are directly exposed to the surrounding inflammatory milieu, thus supporting local immune responses and the production/diffusion of neurotoxic factors.

MS arises from a complex interplay between genetic susceptibility and environmental factors among which Epstein-Barr virus (EBV) has been compellingly related to the risk of developing the disease [8,9].

EBV is a double-stranded DNA herpesvirus, infecting nearly the entire human population, and primary infection is usually asymptomatic in childhood. However, EBV is also the etiological agent of infectious mononucleosis and several lymphoproliferative, malignant disorders [10,11].

EBV mainly infects B lymphocytes and epithelial cells establishing a lifelong latent infection in the memory B cell pool. In the infected cells, EBV can establish four different latency programs (latency 0 to III), defined by the expression of different genes. Upon infection (latency III), all latent proteins are expressed, including EBV nuclear antigens (EBNAs) and latent membrane proteins (LMPs), and their expression is progressively downregulated as the infected cell differentiates into the memory stage [10]. During latency, lytic genes are silenced and the viral episome replicates together with the host genome, so the virus stays hidden from the immune system [12].

EBNA2 is one of the first EBV genes expressed after infection, and functions as a key transactivator of both viral and host genes, promoting the transformation of B lymphocytes into proliferating lymphoblasts [13]. Being the EBNA2 antigen highly immunogenic, cells expressing it can be detected and eliminated by the immune system. To establish long-term persistence, the virus shifts to more restricted latency programs, characterized by minimal viral gene expression.

Under certain triggers—such as co-infections, immune suppression, or pharmacological stimulation—the virus can shift from a latent to a lytic, productive infection, marked by a sequential cascade of gene expression that unfolds in three temporal and functional phases: immediate early (IE), early (E) and late (L). The IE transcription factors BZLF1 and BRLF1 regulate EBV reactivation and activate E genes required for viral DNA replication, while L genes such as *BcLF1* and *BLLF* encode structural components, including the major capsid antigen and the glycoproteins 350/220 (gp350/220), essential for the development of new viral particles [14,15].

After initial infection, both innate and adaptive immune responses (primarily NK cells and CD8^+^ cytotoxic T cells), although failing to eliminate the virus, limit viral replication, allowing for lifelong control of infection and enabling mutual coexistence of the virus and its host. However, disruption of this balance may result in loss of immune control over EBV, permitting viral reactivation and accumulation of EBV-transformed B cells, which can cause inflammatory events and associated pathologies [16]. During IE and E, and, to a lesser extent, during the L phase of the lytic cycle, EBV expresses multiple viral proteins, known as immunoevasins, which inhibit the expression of Toll-like receptor (TLR) signaling, antigen presentation, apoptosis, and interferon responses to evade immune surveillance (Table 1).

So that, as the lytic cycle progresses, epitope presentation efficiency decreases, leading to weaker T cell responses [44].

Traditionally, latent and lytic viral cycles are considered two mutually exclusive mechanisms contributing to EBV persistence. However, in some conditions, infected cells can co-express latent and IE/E lytic gene products in the absence of L lytic gene expression in a process named abortive lytic cycle, reviewed in [45]. This process enables EBV to manipulate its microenvironment to support B cell survival and proliferation and establishing a full-blown latency without completing the replication cycle. EBNA2 and several immunomodulatory and anti-apoptotic EBV proteins expressed during IE and E phases (Table 1) play key roles in dampening antiviral reaction during this process.

The inhibitory immune checkpoint programmed cell death ligand (PD-L)-1 represents one of the major immune evasion strategies employed by EBV. PD-L1 is an inducible membrane protein expressed on the surface of infected or tumor cells that binds its receptor PD-1 on activated T lymphocytes or NK cells, suppressing their cytotoxic and effector functions [46,47]. EBV induces PD-L1 expression through its latent proteins, including EBNA2 [17,20] and LMP1 [48], as well as during the lytic phase of infection [49]. The ability of EBV to activate this pathway in both latent and lytic stages highlights a strategy by which the virus maintains immune privilege and ensures a long-term persistence of the infected B cell populations. Under physiological conditions, in lymphoid organs, the PD-1/PD-L1 axis is involved in regulating germinal center reaction by influencing B cell survival and clonal expansion through interactions between PD-L1-expressing FDC and/or stromal cells and PD-1^+^ CD4^+^ follicular helper T (Tfh) cells [50].

The association between MS and EBV has been widely documented from both epidemiological and immunological perspectives [51], and two main hypotheses have been proposed to explain the underlying mechanisms [52]. EBV may act as a trigger or as a driver of the pathology. The first view implies that EBV infection activates autoreactive B cells or induces molecular mimicry against CNS proteins. However, no definitive MS-linked autoantigens have been identified to date [53]. The EBV driver hypothesis proposes that CNS tissue damage results from a detrimental immune response triggered by a persistent and dysregulated EBV infection in the MS brain, as supported by multiple studies converging on an inefficient immune control of the virus [11,54].

Consistent with the latter hypothesis, EBV DNA [55], RNA [56], and EBV-infected B cells have been observed in the brain lesions and inflamed meninges of SPMS donors, but not in other inflammatory CNS diseases [33,41,57,58,59,60,61,62]. EBV-specific CD8^+^ T cells have been detected in the cerebrospinal fluid (CSF) and brain tissues of people with progressive MS, indicating a local antiviral immune response [41,63,64]. Moreover, reactivation of EBV lytic cycle is increasingly associated with MS; key studies demonstrated EBV lytic gene and protein expression in MS brain tissue, as well as in peripheral blood and CSF [33,55,56,57,61,65,66]; lytic gene expression has also been detected in spontaneous EBV-positive lymphoblastoid cell lines derived from people with MS during active disease [67]. Notably, expansions of EBV-specific CD4 and CD8 T cells have been reported in association with active disease at the radiological and clinical level, suggesting a link between the activation of anti-EBV immunity and CNS inflammation [33,68].

Although cytotoxic lymphocytes, including EBV-specific CD8 T cells, normally limit EBV lytic replication and prevent EBV-driven pathologies, several studies reported that in MS their function becomes progressively impaired, with reduced IFNγ production and cytotoxic activity, consistent with an exhausted phenotype. This inadequate viral control would contribute to sustain neuroinflammation, reviewed in [11].

Meningeal TLSs represent the main intracerebral reservoirs of EBV-infected B cells in MS brains during the progressive phase of the disease, apparently inaccessible to the immune system [4,57,69]. Interestingly, recent data indicate that EBV infection promotes B cell migration and homing to the CNS, suggesting that infected B cells themselves could be the trigger for the formation of TLSs in MS brains [70].

The establishment of immune evasion mechanisms within TLSs [4], together with a dysfunctional antiviral immune response [69] may underlie the persistent, unsolved EBV infection in MS brains. The constantly induced, but ineffective immune response would cause the bystander tissue damage.

In a previous paper [4] we reported the results obtained by performing a detailed immunohistochemical study of the expression and localization of PD-L1 and its receptor PD-1 in post-mortem brain samples from cases with progressive MS and non-neurological controls. We demonstrated that in the cerebral immune infiltrates of MS donors (i) PD-L1 is expressed on EBV latently infected LMP2A^+^ B lymphocytes, mainly accumulating in meningeal TLSs; (ii) T cells and, most importantly, EBV-specific CD8^+^ T cells express PD-1; and (iii) PD-1^+^ T cells establish close contacts with PD-L1^+^/LMP2A^+^ latently infected cells. Neither EBV-infected cells nor PD-L1^+^ cells were detected in any of the non-neurological control brains analyzed. These findings strongly suggested the engagement of the PD-1/PD-L1 pathway in favoring EBV persistence in MS brains.

Since several studies in cell lines and in EBV-associated tumor tissues have reported that PD-L1 expression/upregulation is also associated with EBV lytic infection [49], we have asked whether, in SPMS brains, the PD-1/PD-L1 inhibitory checkpoint is also engaged during the different phases of EBV reactivation. To address this issue, in this study, we have investigated, in SPMS brain tissue, the presence and localization of PD-L1^+^ cells co-expressing EBNA2, an antigen expressed mainly, but not exclusively, during latency III of the viral life cycle, and five EBV antigens of the lytic cycle, namely BZLF1, BHRF1, BALF2, BMRF1, and Gp350/220.

Brain tissue sections from donors who died from non-neurological diseases were used in parallel as neural control samples. Non-pathological lymphoid tissues and tonsil samples from cases with EBV-associated infectious mononucleosis (IM) were used as controls for the immunostainings of the selected cellular and viral markers, respectively.

Finally, given the emerging pathological role of TLSs in SPMS, we have also investigated whether PD-L1^+^ FDC and/or stromal cells, through interactions with PD-1^+^ CD4^+^ T cells (putatively Tfh-like cells) create an immunosuppressive microenvironment that supports TLS formation, organization, and persistence, as well as the maintenance of an intracerebral EBV-infected B cell reservoir.

## 2. Results

### 2.1. EBNA2-Positive B Cells in MS Inflammatory Infiltrates Co-Express PD-L1, and Their Number Positively Correlates with Intracerebral PD-L1 Expression

We analyzed the presence and distribution of PD-L1- and EBNA2-expressing cells, as well as of cells co-expressing both markers, in SPMS and non-neurological control brains. PD-L1^+^ cells were observed in all MS brains analyzed, localized within meningeal and perivascular immune infiltrates of active WM lesions and in the active areas of chronic active ones, as previously described [4]. In line with that already described [57], EBNA-2-expressing B cells were consistently observed in the MS immune infiltrates. Staining performed in serial sections revealed that EBNA2^+^ cells mainly accumulated in the MS immune infiltrates where PD-L1 expression was detected (Figure 1A–F). Notably, in TLSs, a substantial proportion (18%, median value) of EBNA2^+^ cells co-expresses the proliferation marker Ki67 (Figure 1G,H), suggesting the presence of newly infected proliferating B cell blasts in the latency III/growth phase of EBV infection. This finding is in line with concomitant observations by Magliozzi’s group [71]. In non-neurological control brain tissue, only sporadic PD-L1^+^ and no EBNA2^+^ cells were detected, according to previous observations [4]. Double-staining revealed that a subset of EBNA2^+^ cells co-expressed PD-L1 in the inflamed meninges, with the highest frequency observed in the TLS (Figure 1I,J); EBNA2^+^ PD-L1^+^ cells were only occasionally observed in the B cell-enriched perivascular cuffs.

A careful cell count of CD20^+^, EBNA2^+^, PD-L1^+^ cells, and of cells co-expressing EBNA2 and CD20 or PD-L1 was performed in five MS cases throughout four entire serial sections/case. We found that EBNA2 is expressed on 7.7% ± 1.5% (median value = 7.5%) of B cells in meningeal TLSs. This percentage was significantly lower in non-aggregated meningeal and in perivascular B cell-enriched infiltrates in active WM lesions (4.5% ± 2.0%, median value = 4.5%, and 4.2% ± 1.1%, median value = 4.8%, respectively; *p*-value < 0.05; Figure 1K). A negligible percentage of B cells (<0.1%) expressed EBNA2 in perivenular immune infiltrates in reactive NAWM, and no double-positive cells were detected in the gray matter, where no B cell-enriched infiltrates were observed.

The percentages of EBNA2^+^ CD20^+^ B cells with respect to the total CD20^+^ B cell population, and of EBNA2^+^ PD-L1^+^ cells with respect to the total number of PD-L1^+^, were calculated in the different brain areas, TLSs, non-aggregated meningeal infiltrates, and perivascular infiltrates in the WM. A positive correlation was found between the percentage of B cells expressing EBNA2 and the number of PD-L1^+^ cells, both in the meningeal TLSs (R^2^ = 0.901, Pearson correlation = 0.95; *p*-value = 0.004) and in sparse meningeal immune infiltrates (R^2^ = 0.831, Pearson correlation = 0.91; *p*-value = 0.003) (Figure 1L,M). In three perivascular B cell-enriched infiltrates in WM active lesions from different MS donors, the Pearson correlation between B cells expressing EBNA2^+^ and the number of PD-L1^+^ cells was 0.66. Given the limited number of observations, these data did not reach statistical significance.

These findings confirm previous data describing the presence of EBNA2^+^ cells in inflammatory infiltrates in WM and meninges of SPMS cases characterized by significant inflammation and ectopic follicle formation [57,71]. Notably, these findings in MS brain tissue are in line with previous observations showing EBNA2-induced upregulation of PD-L1 in EBV-infected tumor cells [17,20].

#### 2.1.1. Presence and Distribution of Cells Expressing the EBV E Antigen BHRF1 in MS Brain

BHRF1 is an EBV gene homologous to Bcl-2, expressed from early infection through the E lytic phase. It is involved in B cell transformation and viral replication and contributes to viral immune evasion [72]. Preliminary experiments were performed in tonsillectomy samples from a child with recurrent bacterial infections and from two individuals with EBV-associated IM, aiming at establishing the appropriate immunohistochemical procedures for BHRF1 detection (Appendix A). We next investigated the presence and distribution of cells expressing this EBV E lytic antigen in SPMS brains.

Immunohistochemical analysis revealed BHRF1 immunoreactivity on several cells in meningeal TLSs and in B cell-enriched WM perivascular infiltrates (Figure 2A,B). According to previous descriptions in vitro [73], BHRF1 immunoreactivity showed both a cytoplasmic and, less frequently, a nuclear intracellular localization. Cells co-expressing BHRF1 and EBNA2 were consistently observed in the meningeal infiltrates (Figure 2C), indicating the presence of recently infected cells and/or of infected cells in the abortive phase of viral reactivation.

#### 2.1.2. Cells Expressing EBNA2 and Early Lytic Antigens BHRF1, BALF2, and BMRF1 Co-Express PD-L1

We next performed double-staining for PD-L1 and five different EBV antigens of the lytic phase. PD-L1 expression was detected on cells expressing three different early lytic antigens: BHRF1 (Figure 2D–J), BMRF1 (Figure 2K), and BALF2 (Figure 2L), while no co-expression of PD-L1 with the IE antigen BZLF1 (Figure 2M) or the L glycoprotein gp350/220 was observed. Cells co-expressing PD-L1 and BHRF1, BMRF1, and BALF2 EBV E lytic antigens were detected in meningeal infiltrates, where EBNA2^+^ and EBNA2^+^/PD-L1^+^ cells were also present, while they were only occasionally observed in the WM infiltrates. Close contacts between cells expressing EBV E lytic antigens and PD-1^+^ lymphocytes were frequently detected in the B cell enriched meningeal infiltrates, suggesting a possible functional interaction between PD-L1 and its receptor (Figure 2N,O).

To investigate whether mechanisms similar to those described in EBV^+^ tumors might operate in MS brain lesions, we next examined the expression of two cytokines involved in PD-L1 regulation and viral immune evasion. IFNγ works synergistically with EBNA2 to induce PD-L1 on EBV^+^ tumor cells [74]; the immunosuppressive cytokine IL-10 acts as critical pathway in suppressing the host’s antiviral T cell response [75]. PD-L1 and IL-10 can work together, often in feedback loops, to repress immune responses in tumors: IL-10 promotes PD-L1 expression on cancer cells, and PD-L1 signaling also leads to IL-10 release. Therefore, we analyzed the expression of these cytokines in the MS cerebral immune infiltrates in serial sections with those where PD-L1^+^ cells have been observed. In all samples analyzed, we found both a prominent IFNγ expression, and the presence of CD4^+^ T cells producing IL-10 within the meningeal TLSs where PD-L1^+^ cells accumulate (Figure 2P,Q).

These observations reinforce the hypothesis that, similarly to that reported in oncogenesis, these cytokines might play a role both in PD-L1 upregulation in MS brains and in establishing and maintaining an immunosuppressive microenvironment.

#### 2.1.3. Quantification of EBV-Infected Cells Co-Expressing PD-L1

Careful cell counts performed on about 80 serial sections from eight tissue blocks of six MS cases, subjected to single- and double-immunostainings, revealed that more than 60% of PD-L1^+^ cells co-expressed the latency EBV antigens EBNA2 and LMP2A in the inflamed meninges (49 ± 5% and 15 ± 3%, respectively). Approximatively 25 ± 8% of total PD-L1^+^ population co-expressed BHRF1 in inflamed meninges, whereas the 9 ± 3% co-expressed BMRF1. The percentage of PD-L1^+^ cells co-expressing BALF2 was more variable across the MS cases analyzed, accounting for 13 ± 10% (Figure 3A). Among cells immunoreactive for the EBV antigen analyzed, LMP2A^+^ and EBNA2^+^ cells showed the highest proportion of co-expression with PD-L1 (60.5 ± 19.5% and 34.5 ± 9%, respectively), with the highest percentage observed in the meningeal TLSs. Regarding E lytic EBV antigens, the highest co-expression of PD-L1 was observed with BHRF1 (56 ± 10%), an antigen also expressed during the EBV latent phase, and about 27% of PD-L1^+^ cells co-express BALF2 or BMRF1, antigens typical of the viral E lytic phase (Figure 3B). These observations suggest an ongoing viral immune evasion activity in SPMS brain inflammatory infiltrates, mainly in meningeal TLSs and in the surrounding areas, occurring not only during latency, but also in the early lytic phase of viral reactivation.

#### 2.1.4. FDCs Express PD-L1, and Intrafollicular CD4^+^ T Cells Express PD-1 Within EBV-Storing Meningeal TLSs

Meningeal TLSs in the SPMS brain have been identified as immunoprivileged niches that favor the accumulation of EBV-infected cells, mainly in the latent phase of infection. To further characterize these ectopic structures and explore the mechanisms allowing for the establishment of a permissive microenvironment for the virus persistence, we tested the hypothesis that, similarly to B follicles in secondary lymphoid tissues, FDC express/upregulate PD-L1 and interact with intrafollicular CD4^+^ T cells expressing PD-1 within meningeal TLSs in MS.

Since studies on EBV^+^ cell lines and tumors described as EBV-encoded small RNAs (EBERs) released by infected B cells can stimulate TLR3 on FDC and induce PD-L1 expression [76,77], we performed, on a serial MS brain section, double-staining for the FDC/stromal cell marker CD35 and PD-L1 and in situ hybridization for EBER. We observed numerous cells co-expressing CD35 and PD-L1 in the meningeal TLSs, where EBV-infected EBER+ cells accumulate (Figure 4A–D). In the same TLSs, PD-1^+^ CD4^+^ T cells were also detected, often in contact with PD-L1^+^ cells with dendritic/stromal morphology (Figure 4E–G).

These findings suggest that, within meningeal TLSs, PD-L1 expressed by FDCs or stromal cells could interact with its receptor on T cells. This interaction can influence local B cell survival (including of EBV-infected cells) and promote clonal expansion, thereby contributing to the establishment of an immune-permissive habitat favoring the maintenance of an intracerebral EBV reservoir.

## 3. Discussion

This work provides neuropathological evidence that EBV uses the PD-1/PD-L1 inhibitory axis during most phases of its life cycle to establish permanent niches of infection in MS brains. This study highlights the involvement of EBNA2 and selected antigens of the EBV E lytic cycle in immune evasion. Moreover, it provides additional insight into TLS organization and their potential contribution to MS chronic neuroinflammation.

In a previous study, we demonstrated the presence in SPMS brains of latently EBV-infected cells co-expressing LMP2A and PD-L1, of EBV-specific cytotoxic CD8^+^ T cells expressing PD-1, and the establishment of close contact among them, suggestive of intracerebral functional interactions, possibly aimed at holding the immune system in check during latent infection. Several studies in cell lines and tumor tissues have reported PD-L1 expression/upregulation also associated with EBV lytic infection [49,78,79,80]. The study of Yanagi and collaborators suggested that PD-L1 upregulation upon lytic induction is linked to the increased expression of EBNA2, and the subsequent activation of specific cellular signaling pathways.

The aim of this study was to perform in post-mortem MS brain tissue, using multiple immunostainings, a detailed analysis of PD-L1 expression, and distribution in EBV-infected cells at different stages of the viral life cycle. This was performed to further investigate the mechanisms underlying EBV-mediated immune evasion.

To this purpose, we analyzed the possible co-expression of PD-L1 and EBNA2 (which is expressed during latency III and, under certain conditions, during the E phase of viral reactivation), and of PD-L1 and five viral gene products related to different stages of the lytic cycle. In addition, the expression of IFNγ—which upregulates PD-L1 on infected cells [81,82]—and of IL-10, which is induced during immune evasion [75], was examined.

Notably, within this investigation, we described for the first time the cellular expression and tissue distribution of the EBV antigen BHRF1 in the brains of SPMS patients. BHRF1 is a viral protein expressed from the early stages of infection through the E lytic phase [72]. It plays a role in viral immune evasion by preventing apoptosis of infected cells and IFNβ activation and by blocking the nuclear translocation of IRF3, a transcription factor that plays a crucial role in innate immunity, particularly in response to viral infectious agents.

The first finding of this study is that in meningeal inflammatory infiltrates, mainly in TLSs, a substantial proportion of EBNA2^+^ cells express PD-L1. Moreover, at these inflammatory sites, a notable percentage of EBNA2^+^ cells co-expresses the proliferation marker Ki67, indicating the presence of proliferating B cell blasts in the latency III/growth phase of the infection, and/or in the E phase of the viral lytic cycle, when EBNA2 is still expressed (abortive lytic cycle). These observations, together with the prominent expression of IFNγ within the TLSs, are consistent with the results arising from previous studies performed on B cell lymphomas and nasopharyngeal carcinoma, demonstrating that EBNA2 induces PD-L1 on EBV infected B cells, acting in synergistic manner with IFNγ [17,74].

Another finding arising from this study is the detection of PD-L1 on a fraction of cells expressing three different EBV antigens of the E lytic cycle, BHRF1, BALF2, and BMRF1, in inflamed meninges and, only rarely, in B cell-enriched perivascular infiltrates. By contrast, no co-expression of PD-L1 and the IE EBV antigen BZLF1 or the gp350/220 L glycoprotein was detected.

These observations suggest that in the MS brain the upregulation of the immune checkpoint molecule PD-L1 is apparently restricted to infected cells expressing antigens related to latency and to the E phase of the EBV lytic cycle.

Moreover, the detection in TLSs of cells expressing IE and E lytic EBV antigens along with EBV-infected cells in the latency III program provides evidence that a localized viral reactivation occurs within intracerebral EBV survival niches. Because cells expressing late antigens were rarely detected, this reactivation process appears to produce few, if any, new viral particles. The first conclusion we can draw from the consistent PD-L1 expression on subsets of EBNA2-positive and of cells expressing E lytic antigens is that the virus itself tightly regulates the productive phase.

As observed in tumors, an abortive lytic cycle may occur also in the SPMS brain during a pre-latent phase of infection to establish a persistent, long term latency. This represents a crucial “third state” of EBV infection, distinct from both latency and the lytic cycle. During this phase, cells co-expressing EBNA2 and certain E lytic antigens are present in the infected tissues and are easily recognizable to the immune system. We can hypothesize that the upregulation of PD-L1 would help these cells to evade the immune response and complete the process until the full latency state is reached. In support of this hypothesis, we identified cells co-expressing EBNA2 and BHRF1 in inflamed meninges that also co-expressed PD-L1.

Therefore, EBV exploits the PD-1/PD-L1 inhibitory pathway to limit excessive local activation of virus-specific memory and effector CD8^+^ T cells: by disrupting B–T cell communication, the virus contributes to T cell dysfunction and exhaustion; PD-L1 expression on regulatory B cells can engage PD-1 on T cells, triggering IL-10 production and further suppressing T cell activity.

The absence or undetectable expression of PD-L1 on cells expressing the BZLF1-encoded antigen may represent a self-regulating mechanism of infection, allowing cytotoxic CD8^+^ T cells to more effectively eliminate infected cells in the IE phase of the lytic cycle, and thereby limit the extent of infection within the host.

The well-established hierarchy of immunodominance among CD8^+^ T cell responses targeting EBV lytic cycle antigens, with the two immediate-early antigens, BRLF1 and BZLF1, eliciting the strongest immune responses [26,44], supports this last hypothesis. In the peripheral blood of patients with relapsing MS, BZLF1 elicits a major cytotoxic immune response during disease flares, and has been implicated in the active phase of the disease [33]; in our cohort of brain samples, EBV-specific CD8T cells recognizing the BZLF1 antigen were more numerous than CD8T cells targeting other EBV antigens [41]; furthermore, a study on MS patients where the presentation of EBV peptides by HLA-E—a non-classical MHC class I molecule that interacts with both NK cell receptors and the TCR of cytotoxic CD8^+^ T cells—was investigated, demonstrated an increased HLA-E–restricted recognition of the BZLF1-derived peptide by CD8^+^ T cells in MS patients compared to healthy controls [83].

However, due to a defective immune control and without apparent involvement of the PD-1/PD-L1 axis, a subset of cells expressing IE antigens can evade immune surveillance, progress to later stages of the lytic cycle—either abortive or productive—and induce local inflammation by promoting the production of highly detrimental cytokines, such as IL-6, IL-8, IL-10, IL-13, and IL-1β [84].

Since significant destruction of host tissue cells—such as that caused by the release of new viral particles in cancer or chronic infections—is not beneficial to the virus, it must closely monitor immune responses, modulate the transcriptional activity of antiviral and pro-inflammatory pathways, and interfere with HLA-mediated antigen recognition. These mechanisms allow for the virus to maintain a steady level of infection, comprising a reservoir of potentially productive infected cells, while avoiding an excessive immune response that could irreparably damage the host (Figure 5).

This study also sheds further light on TLSs organization and their potential contribution to the neuroinflammation in SPMS. Within TLSs, where EBER^+^ cells were present, beyond on EBV-infected B lymphocytes, PD-L1 immunoreactivity was also detected on CD35^+^ FDCs, and were frequently observed intrafollicular CD4^+^ T cells expressing PD-1. Notably, PD-L1^+^ CD35^+^ FDCs and PD-1^+^ CD4^+^ T cells—presumably Tfh cells—were often found in close contact. We hypothesize that within meningeal TLSs, as described in EBV^+^ cell lines and tumors [76], EBER particles released from EBV-infected cells, often via exosomes, could act as a ligands to activate TLR3 on FDC and neighboring stromal cells. This would trigger the induction on these cell types of PD-L1, and a downstream inflammatory antiviral response, including the production of type I interferons. In TLSs, PD-L1 could act both as an “organizer” of germinal centers (as in secondary lymphoid organs) and as an inhibitor of the effector functions; hence, in MS inflamed meninges, through pathogenic events mediated by the EBER-TLR3/FDC-PD-L1 axis, the virus itself may promote the development of TLSs, using them as a protected reservoir of infected cells.

These results add new insights into possible pathogenic mechanisms organizing and sustaining a persistent, poorly controlled, intracerebral EBV infection in SPMS.

## 4. Materials and Methods

### 4.1. Tissue Samples

Human postmortem brain tissues from donors with progressive MS and non-neurological control cases have been analyzed in this study. All brain samples were provided by the UK Multiple Sclerosis Tissue Bank at Imperial College London (https://www.imperial.ac.uk/medicine/multiple-sclerosis-and-parkinsons-tissue-bank/, assessed on 10 December 2025) with fully informed consent via a prospective donor scheme. Based on the available clinical histories, all MS cases were in the progressive phase of the disease and were wheelchair- or bed-bound at the time of death (Expanded Disability Status Scale > 7). No treatment was reported during the progressive MS phase. For MS cases, mean disease duration was 17 + 9 years and mean age at death was 43 + 7 years. Donor information is provided in Table 2.

Control lymphoid tissues included one abdominal and one axillary autopsy lymph node from a control subject (fixed in 4% paraformaldehyde and then frozen: FF samples), and paraffin samples from one tonsil from a child with recurrent bacterial infections undergoing tonsillectomy (provided by the Institute of Pathological Anatomy, U.C.S.C. Policlinico A. Gemelli, Rome, Italy), and from two tonsils from two cases of EBV-associated infectious mononucleosis (IM; kindly provided by Prof. Gerard Niedobitek, Institute of Pathology, Sana Klinikum Lichtenberg, Berlin, Germany). Lymph nodes and non-infected tonsils were used as internal controls of the immunostainings for immune cell markers (B and T lymphocytes, FDCs, Ki67), PD-1, and PD-L1; IM samples have been used as control for immunostainings of EBV markers.

### 4.2. Neuropathological Assessment

After extensive neuropathological screening performed on brain tissue blocks from more than twenty MS donors did, well-preserved cerebral tissue blocks with substantial B cell infiltration and presence of intrameningeal ectopic tertiary lymphoid structures (TLSs) were selected. Thirteen tissue blocks (4 cm^3^) from superior frontal gyrus, precentral gyrus and middle temporal gyrus from eight MS cases were analyzed. Eight brain blocks from seven MS cases were fixed in 4% paraformaldehyde (PFA) and frozen (FF samples) and stored at −80 °C until use, four brain blocks from four MS cases were snap-frozen (SF samples), and one block from one MS donor was formalin-fixed and paraffin-embedded (FFPE). All brain blocks from control cases were without neurological disease, and the lymph nodes were FF and three tonsils were FFPE samples (Table 2).

Serial 10 µm and 4 µm thick sections were cut from the selected FF or SF and FFPE tissue blocks, respectively. Neuropathological features of MS tissues (extent of demyelination, lesion inflammatory activity, degree of immune cell infiltration in WM, and meninges) and lymphocyte composition of immune infiltrates were assessed on the first sections of each series by histological and immunohistochemical staining, as previously described [41,69]. Substantial B and T cell cerebral infiltration and presence of meningeal B cell follicles were detected in all MS tissue blocks selected and analyzed. Negligible or no meningeal and parenchymal immune infiltration was detected in control brains. Subsequent sections of each series and sections from tonsils and lymph nodes were analyzed using immunohistochemical staining for the selected viral and cell markers.

### 4.3. Immunohistochemistry

SF and FF sections were air-dried for two hours at room temperature (RT) and then fixed in cold acetone for 10 min at 4 °C and then air-dried for 20 min at RT. FFPE sections were dewaxed in xylene and rehydrated using a decreasing ethanol series. After rehydration in PBS, FF and FFPE sections were subjected to the appropriate antigen retrieval treatment. Antibodies, staining conditions, and antigen retrieval treatments are shown in Table 3.

Single-immunostaining in the bright field was performed using mouse monoclonal antibodies (mAbs) specific for CD20, CD35, PD-1, PD-L1, EBNA2, BALF-2, BZLF-1, BMRF-1, and gp250/350, rabbit mAbs specific for BHRF1, CD4, PD-L1, and IFNγ, and rat mAbs specific for EBNA2 and IL-10. Primary antibody binding was visualized using biotin-conjugated rabbit anti-mouse (Thermo Fisher Scientific, Waltham, MA, USA), goat anti-rabbit, or donkey anti-rat immunoglobulin (Ig) (Jackson ImmunoResearch Laboratories, Cambridgeshire, UK), avidin-biotin complex (ABC) (Vector Laboratories Inc., Burlingame, CA, USA), and 3,3′-diaminobenzidine tetrahydrochloride (DAB) (Sigma Aldrich, St. Louis, MO, USA) or 3-Amino-9-ethylcarbazole (AEC) as chromogens. Double-staining in the bright field and/or immunofluorescence were performed using combinations of antibodies against the following: CD20 and PD-L1 or EBNA2; PD-L1 and different EBV antigens (EBNA2, BHRF1, BALF-2, BZLF-1, BMRF-1 and gp250/350) or CD35; BHRF1 and EBNA2; PD-1 and BHRF1 or CD4; CD4 and IL-10 or CD35; and EBNA2 and KI67.

For double-staining in the bright field, the Zytochem plus two-step double-stain polymer kit (Zytomed Systems GmbH, Berlin, Germany) was used, as previously described [69]. Briefly, after incubation with a combination of rabbit and mouse primary antibodies, sections were treated sequentially with AP-Polymer anti-rabbit, Tris-buffered saline (TBS), permanent AP Red Kit containing Levamisol (Vector Lab) or Ferangi Blue™ Chromogen Kit 2 (Biocare Medical, Pacheco, CA, USA), TBS, horseradish peroxidase (HRP)-conjugated anti-mouse polymer, and permanent HRP Green Kit (green/blue color; Zytomed System) or DAB. When DAB was used for detection, the sequence of steps was inverted: DAB was applied first, followed by the permanent AP Red Kit or Ferangi Blue chromogens. After very quick washing in distilled water, sections were sealed with Ultramount Aqueous permanent mounting medium (Agilent Dako, Santa Clara, CA, USA) and analyzed with an Axiophot microscope (Carl Zeiss, Jena, Germany) equipped with AxioCam MRc5 camera and Axiovision SE 64 software. Double-immunofluorescence was performed using the appropriate combination of the fluorophore-conjugated secondary antibodies (2 µg/mL; Thermo Fisher Scientific and Jackson Laboratories) and sealed with Prolong Gold antifade reagent with DAPI (Thermo Fisher Scientific) as described [33]. Sections were finally analyzed with a Zeiss Axioscope epiluminescence microscope equipped with an Axiocam 512 digital camera and ZEN 3 Lite software.

### 4.4. Cell Counts and Statistical Analysis

Cell counts were performed manually through the entire sections by two researchers in an independent manner using a morphometric grid and 20× and 40× objectives. Thirteen meningeal TLSs, inflamed meninges containing non-aggregated lymphocytic infiltrates and WM perivascular cuffs, were analyzed. The mean area of the meningeal TLSs was 36,409 ± 21,090 µm^2^, median value = 37,432 µm^2^. CD20^+^ B cells, PD-L1^+^ cells, cells positive for specific EBV antigens (EBNA2, BZLF1, BHRF1, BALF2, BMRF1, GP350/250), and cells co-expressing PD-L1 and each EBV-specific antigen analyzed were counted in at least 10 serial sections from 11 tissue blocks of 8 MS cases; CD20^+^/EBNA2^+^ cells were counted in serial sections from 6 tissue blocks of 6 MS cases. The percentage of cells co-expressing CD20 and EBNA2 was calculated relatively to the total CD20^+^ B cell population and correlated to the number of PD-L1^+^ cells counted in the same microscopic field. The percentages of cells co-expressing PD-L1 and the single specific EBV antigens were calculated both relatively to the total PD-L1^+^ cell population and to the total number of cells expressing the single EBV antigens. Comparisons between groups (e.g., TLSs, non-aggregated meningeal infiltrates, and perivascular cuffs) were performed using the two-tailed unpaired Student’s *t*-test. Correlations between the percentage of EBNA2^+^ B cells and the number of PD-L1^+^ cells in different groups were evaluated using Pearson correlation coefficient (r), and the coefficient of determination (R^2^) was reported. A *p*-value < 0.05 was considered statistically significant. Quantitative data were expressed as mean ± standard deviation (SD).

## 5. Conclusions

This is the first study describing, in MS brain tissue, the expression of PD-L1 on EBV-infected cells in different stages of the virus life cycle. Previous studies exploring this topic examined EBV-positive and -negative cell lines, EBV^+^ LCL, or EBV-associated tumor tissues.

Our results highlight that EBNA2, EBV E lytic antigens, and PD-L1 expression seem to work in synergy to facilitate viral persistence in MS brains, which fuels a sustained and ineffective but detrimental inflammatory immune response, provoking tissue damage and disease progression.

Moreover, this study offers new data into the factors driving TLSs formation/organization in MS brains, reinforcing the concept that intrameningeal TLSs, with their intricate network of immune regulation (also driven by the virus itself), are the ideal microenvironment for EBV persistence.

A limitation of this study is the limited number of brain samples analyzed. This was due to the extensive neuropathological screening required to identify suitable tissue blocks. Since TLSs—the main EBV reservoir in the MS brain—have not been identified in cerebral tissue from donors at earlier disease stages, we focused our analysis on tissues donated by individuals with SPMS.

Understanding the complex interplay between the virus and the host’s immune response and defining the dynamics and variability of EBV lytic reactivation processes in MS brains is essential for developing targeted therapies that can modulate the immune response by disrupting the mechanisms driving chronic intracerebral inflammation. This knowledge also provides a further theoretical basis for designing EBV vaccines and antiviral treatments.

## Figures and Tables

**Figure 1 ijms-27-00437-f001:**
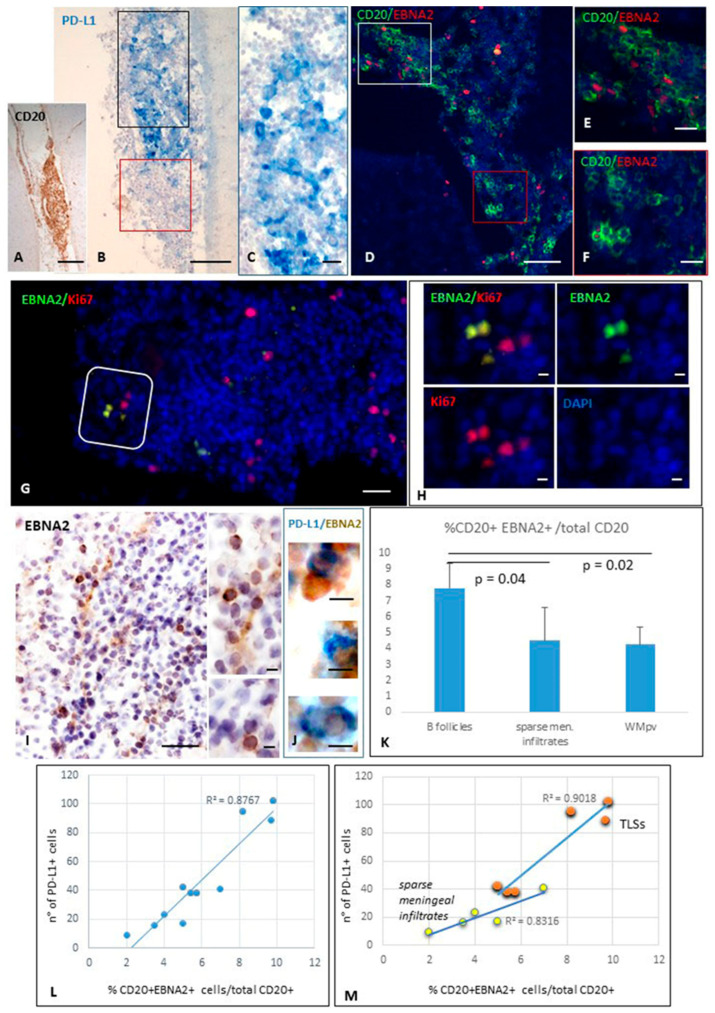
Expression of PD-L1 and EBNA2 in TLSs forming in meninges of SPMS brain. Immunohistochemistry for CD20 (**A**) and PD-L1 (**B**,**C**) and double-immunofluorescence staining for CD20 (green) and EBNA2 (red, **D**–**F**) performed on serial brain sections show the presence of PD-L1^+^ and EBNA2^+^ B cells within the same intrameningeal B cell follicle. Panel (**C**) shows at high magnification the area comprised in the black square in (**B**); panel (**E**) shows at high magnification the immune cells enclosed in the white square in (**D**); panel (**F**) is an high magnification of the areas comprised in the red squares in (**B**,**D**), where both PD-L1^+^ and EBNA2^+^ cells are not present. Double-immunostaining for EBNA2 (green) and KI67 (red; **G**,**H**) reveals the presence of EBNA2^+^ cells co-expressing Ki67. In a meningeal infiltrate where EBNA2^+^ cells were identified (**I**), double-bright-field immunohistochemistry (**J**) demonstrates the presence of EBNA2^+^ cells (brown) co-expressing PD-L1 (blue). (**K**) Histogram showing the proportion of CD20^+^ EBNA2^+^ cells among total CD20^+^ B cells. EBNA2 expression was significantly higher in TLS B cells than in non-aggregated meningeal and perivascular infiltrates. Bars represent mean +/− SD. Statistical significance was assessed by Student’s *t*-test. (**L**,**M**) Scatter plots showing the positive correlation between the percentage of EBNA2^+^ B cells and the number of PD-L1^+^ cells, as assessed by Pearson’s correlation coefficient (r), in inflamed meninges (**L**) (R^2^ = 0.8767, r = 0.94; *p*-value < 0.001), in meningeal TLSs (R^2^ = 0.901, r = 0.95, *p*-value = 0.004) and in sparse meningeal infiltrates (R^2^ = 0.831, r = 0.91, *p*-value = 0.003) (M). Bars: 100 μm in (**A**,**B**,**I**); 50 μm in (**D**,**G**); 20 μm in (**C**,**E**,**F**); 10 μm in (**H**,**J**) and the inset in (**I**).

**Figure 2 ijms-27-00437-f002:**
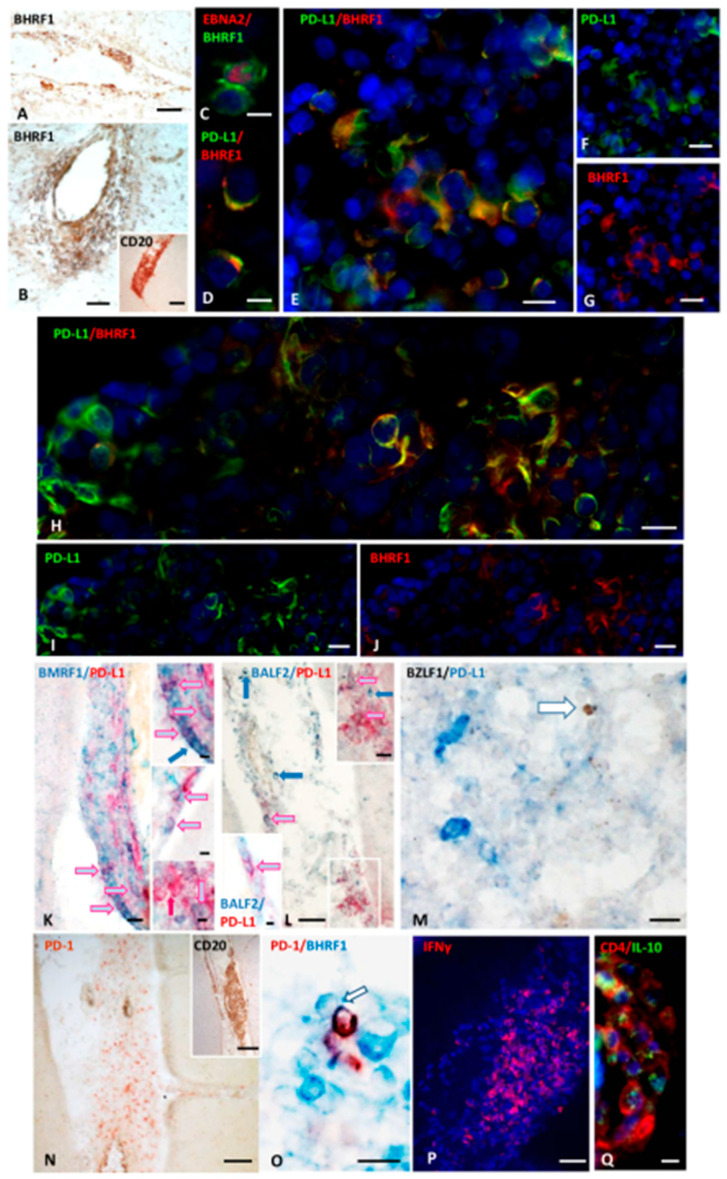
**BHRF1 immunoreactivity, and co-expression of PD-L1 and Early lytic EBV antigens in inflamed meninges of SPMS donors.** Immunohistochemical staining for BHRF1 (**A**,**B**) and CD20 (inset in (**B**)) show the presence of BHRF1-expressing cells in a meningeal infiltrate and in a B cell–enriched perivascular cuff in WM, respectively. Double-immunofluorescence staining for EBNA2 (red) and BHRF1 (green; (**C**)), reveals the presence of BHRF1^+^ cells co-expressing EBNA2 in a meningeal infiltrate. Double-immunostaining for PD-L1 (green) and BHRF1 (red; (**D**–**J**)), shows co-expression of BHRF1 and PD-L1 on some cells within two intrameningeal TLSs from two SPMS cases (**D**–**J**). Panel (**D**) highlights at high magnification the preferential cytoplasmic and membrane distribution of BHRF1 and PD-L1, respectively. Bright-field double-immunohistochemical staining for BMRF1 (blue) and PD-L1 (red; (**K**) and insets) and for BALF2 (blue) and PD-L1 (red; (**L**) and insets) show the presence of PD-L1^+^ cells co-expressing BMRF1 and BALF2 (pink/pale blue arrows) in TLSs (**K**) and meningeal infiltrates (**L**). The upper inset in (**K**) shows at higher magnification the cells indicated by the arrows in (**K**); blue arrows point to three BMRF1^+^/PD-L1^− ^cells; red arrow in the lower inset indicates one PD-L1^+^ BMRF1^−^ cell. The upper inset in (**L**) shows at higher magnification the area in the white square in (**L**); the pink/pale blue arrow in the lower inset points to one BALF2^+^/PD-L1^+^ cells in a meningeal infiltrate. Double-staining in bright field for BZLF1 (brown) and PD-L1 (blue; (**M**)) shows no co-expression of PD-L1 and the EBV IE antigen BZLF1 in a WM lesion. Immunohistochemistry for PD-1 (**N**) and CD20 (inset in (**N**)), and double-staining for PD-1 (brown) and BHRF1 (blue; (**O**)) on serial brain sections show the presence of PD-1^+^ lymphocytes in close contact with BHRF1^+^ cells. In the same inflammatory infiltrate an intense IFNγ production (red; (**P**)) and CD4^+^ cells (red) expressing IL-10 (green; (**Q**)) are also present. Bars: 100 μm in (**N**) and the insets in (**B**,**N**), 50 μm in (**A**,**B**,**L**,**P**); 20 μm in (**E**–**K**,**M**,**Q**); 10 μm in (**C**,**D**,**O**), and in the insets in (**K**,**L**).

**Figure 3 ijms-27-00437-f003:**
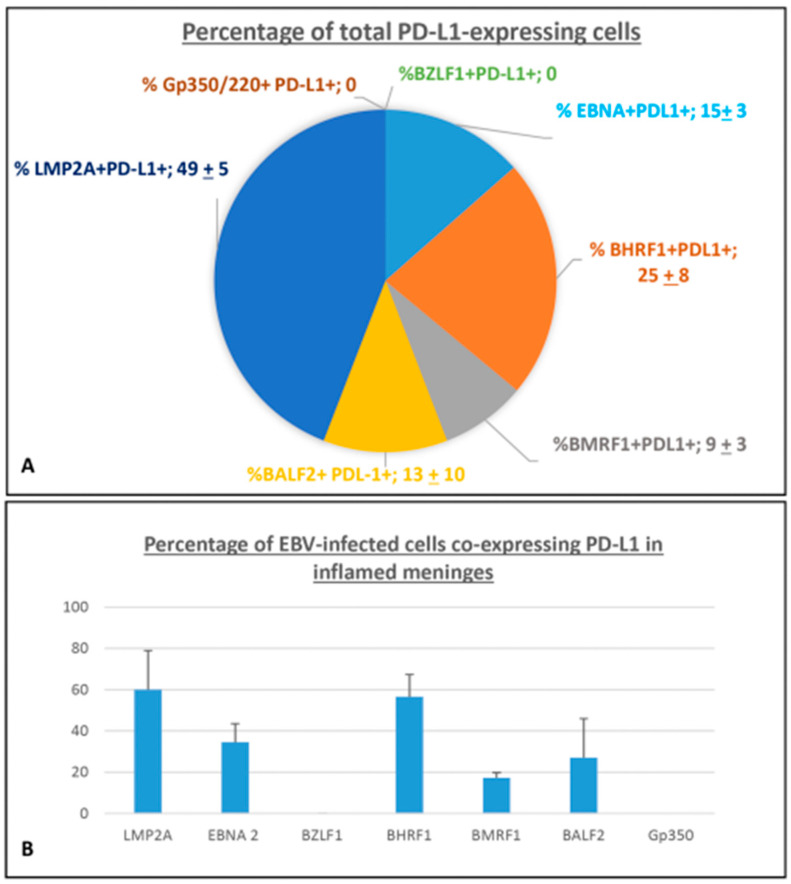
**Percentage of cell expressing PD-L1 in inflamed meninges of SPMS donors.** (**A**) The pie chart shows the percentage contribution of each EBV antigen analyzed to the total population of cells expressing PD-L1 in the inflamed meninges of SPMS brains. Each slice represents mean ± SD. (**B**) The histogram shows the percentage of cells co-expressing PD-L1 within different cell populations expressing single-EBV antigens in inflamed meninges. Bars represent represents mean ± SD.

**Figure 4 ijms-27-00437-f004:**
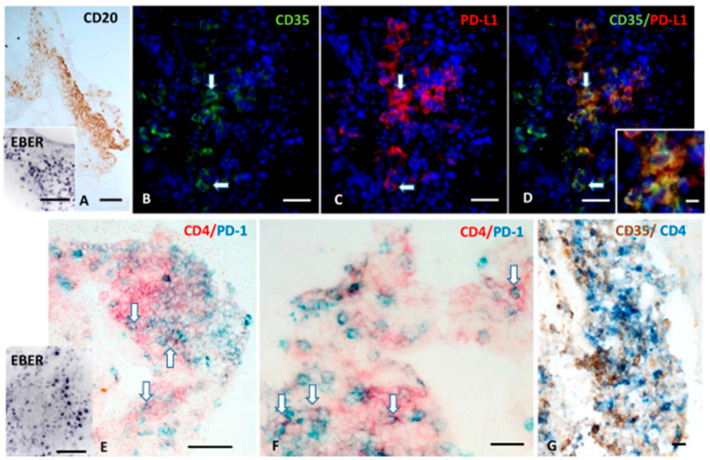
**FDC express PD-L1 and are in close contact with intrafollicular CD4^+^ T cells expressing PD-1 within a meningeal TLSs.** CD20 immunohistochemistry and in situ hybridization for EBER (**A** and inset in **A**) shows a TLS containing several EBV-infected cells forming in the inflamed meninges of a MS donor. Double-immunofluorescence for CD35 (green) and PD-L1 (red; **B**–**D**) performed in a serial section shows the presence of CD35^+^ cells expressing PD-L1 (orange, arrows). Double-immunohistochemistry for CD4 (red) and PD-1 (blue; **E**,**F**) and in situ hybridization for EBER (inset in **E**) performed in a serial section reveals the presence of CD4^+^ T-cells expressing PD-1 in the same meningeal infiltrate shown in (**A**), where several EBV-infected cells are consistently present. Double immunostaining for CD35 (brown) and CD4 (blue; **G**) in a third serial section shows the establishment of close contact between PD-1^+^ CD4^+^ T cells and CD35^+^ FDC. Bars: 100 μm in (**A**) and in the insets in (**A**,**E**); 50 μm in (**B**–**E**); 20 μm in (**F**,**G**); 10 μm in the inset in (**D**).

**Figure 5 ijms-27-00437-f005:**
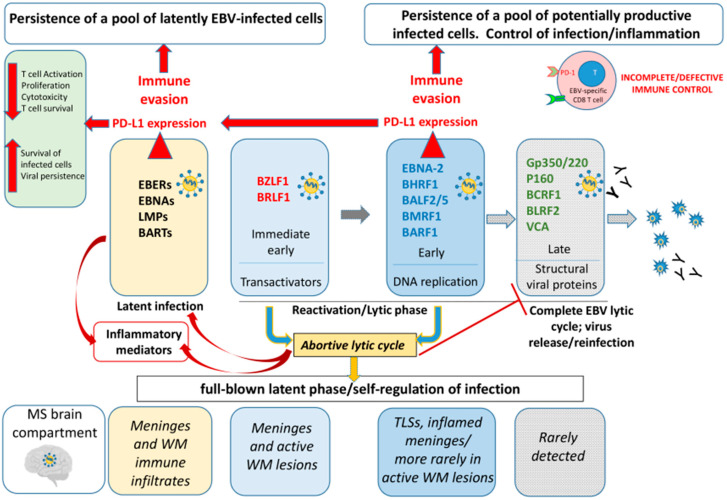
**Hypothetical diagram of the complex mechanisms of EBV persistence in the SPMS brain.** Epstein–Barr virus (EBV) establishes latent infection in the MS brain, primarily within B cells located in tertiary lymphoid structures (TLS) and perivascular infiltrates in active lesions. During latency, EBV upregulates PD-L1 expression (red arrows) to maintain infection while limiting excessive cytotoxic damage to the host (pale yellow boxes). Periodically, EBV can reactivate in the brain, triggering the transcription of immediate early (IE) and early (E) genes involved in viral DNA synthesis and replication (pale blue and blue boxes). This process typically leads to the late (L) phase, when structural viral proteins are assembled and new virions can be produced (gray boxes) with consequent activation of the antibody-mediated immune response. However, in MS brain tissue, antigens associated with the L phase are rarely detected. This indicates that EBV often enters an abortive lytic cycle (yellow box), in which the virus initiates the lytic program, but fails to complete it due to the inhibition of late gene expression. During the E phase of reactivation, in addition to latency, EBV induces PD-L1 expression on infected cells. This activation of an inhibitory immune checkpoint likely serves several purposes: (i) it supports the persistence of both latently infected and potentially productive cells; (ii) by maintaining the abortive lytic state, it allows for long-term infection without producing viral particles, thereby avoiding humoral immune recognition; and (iii) it prevents excessive immune activation that could cause irreversible damage to the host tissue. Overall, PD-L1 expression on EBV-infected cells in the MS brain—during both latent and abortive lytic phases—enables EBV to preserve a stable reservoir of infected cells while minimizing host tissue damage and immune clearance, and functions as a self-regulatory mechanism.

**Table 1 ijms-27-00437-t001:** EBV manipulation of cellular processes to dampen the host immune response during the lytic cycle.

EBV Latency III Gene/Protein	Function	Role in Immune Evasion	Detection in MS Brain	References
EBNA2	Transactivator of viral and cellular genes; it regulates the viral transcription of latency genes and host cell genes; mediates B cell immortalization.	Inhibits the transcription of microRNA-34a (miR-34), increasing the expression of PD-L1; induces miR-24 to reduce ICOSL expression in tumors; impairs MHC-II transcription; creates an anti-inflammatory setting inducing IL-18 receptor on B cells	Yes	[17,18,19,20]
**EBV immediate early lytic cycle** **Gene/Protein**				
BZLF1/transcription factor Z	Starter of the EBV lytic cycle, induces the synthesis of E lytic viral proteins; encodes the transcription factor Z, one of the promoters of early (E) lytic genes that encode the DNA viral replication proteins; mitogen activity.	Interferes with the secretion of IFN-α mediated by the JAK/STAT signaling pathway; inhibits the activation of Interferon Regulatory Factor (IRF7); inhibits the release of inflammatory factors TNF-α and IFN-γ; prevents NF-κB activation.	Yes	[21,22,23,24]
BRLF1/RTA protein	Viral transactivator, permits an ordered cascade of viral gene expression	Suppresses inflammasome activation (specifically RIG-1) and antiviral responses in infected cells; by interacting with subunits of RNA polymerase III suppresses the transcription of viral and cellular RNAs that can be detected by the host’s immune system. Induces BARF1 production	n.r.	[16,25]
**EBV early lytic cycle Gene/Protein**				
BNLF2a	Immunoevasin	It inhibits the transporter associated with antigen processing, blocking antigen presentation to T cells and preventing immune recognition of infected cells.	n.r.	[26,27]
BARF1	Oncogenic, mitogen and immortalizing activity in human epithelial cells.	By blocking CSF-1’s normal function, BARF1 inhibits the development and activation of mononuclear cells, reduces IFN-α release, and hinders NK cell activation and cytotoxicity; up-regulates anti-apoptotic Bcl-2.	Intrathecal Ab synthesis	[28]
BHRF1/vBcl-2 protein	Inhibition of apoptosis; cell survival	It gives rise to viral miRNAs which suppress interferon production and target immune-related genes like CXCL-11 and RIG-I, a key viral sensor; blocking the nuclear translocation of Interferon Regulatory Factor 3, protects BZLF1-sensitized cells from NK cell killing; inhibits apoptosis.	Yes	[29,30]
BMRF1/DNA polymerase processivity factor (PPF)	Viral genome replication and activation of EBV genes. It functions as viral EBV DNA polymerase accessory protein; plays a role in transcriptional activation of some EBV genes for late lytic protein synthesis.	Inactivating the host’s DNA damage response pathway, suppresses the signaling cascade at double-strand DNA breaks, thus inhibiting immune surveillance	Yes	[31]
BFRF1	One of the two essential proteins of the core nuclear egress complex (NEC, with BALF2), essential for the anchoring of the viral capsid, recruitment of factors to reorganize the inner nuclear membrane to allow the viral capsid to exit the nucleus into the cytoplasm.	Suppresses the host’s type I interferon (IFN-I) response, a crucial part of innate antiviral immunity, by blocking the activation pathway of the IRF3 transcription factor.	Yes	[32,33]
BALF2/p138	Component of the NEC. Single stranded DNA binding protein; present in the tegument layer of mature virions. Possible role in both DNA replication and virion assembly.	Prevents host programmed cell death, contributing to maintaining the viral infection within host cells.	Yes	[34]
BGLF5	Viral effector of global mRNA degradation resulting in a severe restriction of cellular gene expression. Genome instability.	EBV-induced host shutoff of host protein synthesis in productively infected cells, resulting in reduced surface display of HLA molecules for T cell recognition. Decreases both RNA and protein expression of TLR9.	n.r.	[35,36,37,38,39]
BMLF1/EB2-SM-Mta	Upregulates expression of GRP78, a central regulator of the unfolded protein response, to maintain host’s cell ER homeostasis and ensure a fully productive replication.	It downregulates BHRF1.	Presence of BMLF1-specific CD8^+^ T cells	[40,41,42]
**EBV Late phase Gene/Protein**				
BCRF1	IL-10 homolog/immunomodulatory protein	Suppression of IFN γ production	n.r.	[43]

n.r.: Not reported.

**Table 2 ijms-27-00437-t002:** Brain donor and sample information.

Case Code	Multiple Sclerosis	Sex/Age at Death (years)	Disease Duration (years)	Cause of Death	Post-Mortem Delay (hours)	Immunotherapy	Tissue Processing	Number of Brain Tissue Blocks Analysed
MS92	SPMS	F/37	17	MS	26	Age 21: ACTH	PFA fixed and frozen	1
							Snap frozen	1
MS180	SPMS	F/44	18	MS	9	Not reported	PFA fixed and frozen	1
							Snap frozen	1
MS234	PRMS	F/39	15	Pneumonia	15	Not reported	PFA fixed and frozen	1
							Snap frozen	1
MS342	SPMS	F/35	5	MS	9	Not reported	PFA fixed and frozen	2
MS352	SPMS	M/43	19	MS	26	Age 32: methylprednisolone Age 33: Campath-1H	PFA fixed and frozen	1
MS408	SPMS	M/39	10	Pneumonia, sepsis	21	Steroids, Mitoxantron, Avonex	PFA fixed and frozen	1
MS121	PRMS	F/49	14	MS	24	Age 46: methylprednisolone	Snap frozen	1
MS330	SPMS	F/59	39	Pneumonia	21	Not reported	PFA fixed and frozen	1
							Formalin fixed-paraffin embedded	1
**Case Code**	**Non Neurological Control**	**Sex/Age at Death** **(years)**	**Disease Duration (years)**	**Cause of Death**	**Post-Mortem Delay (hours)**	**Immunotherapy**	**Tissue Processing**	**Number of Tissue Blocks Analysed**
C14	Signs of ischaemia	M/64		Myocardial infarction	18		PFA fixed and frozen	1
C16	None	M/92		Cardiac failure/old age	13		PFA fixed and frozen	1
C32	Age-related changes	M/88		Prostate cancer	22		PFA fixed and frozen	1
C41	WM and perivascular oedema; mild inflammation	M/54		Lung cancer	20		PFA fixed and frozen	1

SPMS = secondary progressive multiple sclerosis; PRMS = progressive relapsing multiple sclerosis; F = female; M = male; PFA = paraformaldehyde.

**Table 3 ijms-27-00437-t003:** Primary antibodies and staining conditions.

**Antigen**	**Specificity**	**Source**	**Host & Clonality**	**Dilution**	**Tissue Processing** **Post-Fixation**	**Antigen Retrieval**
CD20	B cells	ScyTeK Laboratories, The Hague, The Netherlands	Mouse monoclonal IgG2a, k (clone L26)	Ready to use	FFPE/FF/SF Acetone	Citrate buffer (for FFPE/FF)
CD4	T cells	Quartett, Berlin, Germany	Rabbit monoclonal IgG (clone QR032)	1:250	FF/SF Acetone	Citrate buffer (for IL-10 co-staining) Tris-EDTA-citrate buffer pH 7.8 (for PD-1, and CD35 co-staining)
CD8	T cells	Thermo Fisher Scientific Rockford, Illinois, USA	Rabbit polyclonal IgG	1:800	FF/SF Acetone	Citrate buffer (for FF)
CD35	Follicular dendritic cells/stromal cells	NeoBiotechnologies, Union City, CA, USA	Mouse monoclonal IgG2a, k (Clone CD35 CR1/6378)	1:300	FF Acetone	Tris-EDTA-citrate buffer (for CD4 co-staining)
PD-1		ScyTeK Laboratories	Mouse monoclonal IgG1, k (clone NAT105)	1:100 (for FFPE) 1:200 (for FF/SF)	FFPE/FF Acetone	Citrate buffer (for BHRF1 co-staining)
PD-L1		Quartett	Rabbit monoclonal IgG (clone QR001)	1:120	FF Acetone	Tris-EDTA-citrate buffer (for CD35, EBNA2, BALF-2, BZLF-1, BMRF-1 and gp250/350 co-staining)
PD-L1		Abcam, Cambridge, UK	Mouse monoclonal IgG2a, k (clone ABM4E54)	1:100	FF Acetone	Citrate buffer (for BHRF1 co-staining)
IFN γ		Abcam	Rabbit polyclonal IgG	1:100	SF Acetone	Citrate buffer
IL-10		Thermo Fisher Scientific	Rat monoclonal IgG, k (clone JES3-9D7)	1:150	FF Acetone	Citrate buffer (for CD4 co-staining)
Ki67	Proliferating cells	Spring Biosience Abcam, Cambridge, UK	Rabbit monoclonal IgG (clone SP6)	1:150	FF Acetone	Tris-EDTA-citrate buffer (for EBNA2 co-staining)
EBNA-2	EBV latency III antigen	Celltech, Torino, Italy	Mouse monoclonal IgG1 (clone PE2)	1:10	FFPE/FF/SF Acetone	Citrate buffer Tris-EDTA-citrate buffer (for PD-L1 and KI67 co-staining)
EBNA-2	EBV latency III antigen	Merck KGaA, Darmstadt, Germania	Rat monoclonal IgG2a, k (clone R3)	1:100	FF Acetone	Citrate buffer (for BHRF1 co-staining)
BZLF-1	EBV immediate-early lytic antigen	Novus Biologicals, Colorado, USA	Mouse monoclonal IgG1 (clone BZ.1)	1:50 (FFPE) 1:100 (FF)	FFPE/FF Acetone	Citrate buffer Tris-EDTA-citrate buffer (for PD-L1 co-staining)
BHRF1	EBV early lytic antigen	Cusabio, Houston, USA	Rabbit polyclonal IgG	1:200	FFPE/FF/SF Acetone	Citrate buffer (for PD-1, PD-L1 and EBNA-2 co-staining)
BALF-2	EBV early lytic antigen	Kind gift of Prof. J.M. Middeldorp	Mouse monoclonal (Clone OT13N2)	1:350	FFPE/FF Acetone	Citrate buffer Tris-EDTA-citrate buff
BMRF-1	EBV early lytic antigen	Kind gift of Prof. J.M. Middeldorp	Mouse monoclonal (clone OT14E2)	1:470	FFPE/FF Acetone	Citrate buffer Tris-EDTA-citrate buffer (for PD-L1 co-staining)
gp250/350	EBV late lytic antigen	Thermo Fisher Scientific	Mouse monoclonal, IgG1 (clone C61H)	1:10	FFPE/FF Acetone	Citrate buffer

FFPE = formalin-fixed paraffin embedded; FF = fixed-frozen; SF = snap frozen.

## Data Availability

The original contributions presented in this study are included in the article/Appendix A. Further inquiries can be directed to the corresponding author.

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
