# Peer review of "EBV Early Lytic Antigens, EBNA2 and PDL-1, in Progressive Multiple Sclerosis Brain: A Coordinated Contribution to Viral Immune Evasion"

_ijms, 2025, doi:10.3390/ijms27010437_

Round 1

Reviewer 1 Report

Comments and Suggestions for Authors

The authors studied expressions of EBV antigens and PD-L1 in human brain from multiple

sclerosis (MS) patients. However, there are lots of issues:

  1. It is very hard to collect normal brain tissues. In order to perform a comparative research in this research, the author used lymph nodes from abdomen and axilla as well as tonsils to be control samples. They should not be suitable controls due to different tissues used between cases and controls. The author should at least explain that it is suitable to use these control samples in this research.
  2. The author need to improve the presentation of data. It is confused for me to read the whole manuscript so that I don’t think I clearly understand this research.
  3. There are no title of sub-section 1 in the results section and Fig. 1 was not correctly displayed so that I couldn’t evaluate data from sub-section 1. Please correct them.
  4. The author made a conclusion based on data “A prominent IFNγ expression and the presence of CD4+T cells producing IL-10 were detected within the meningeal TLSs where PD-L1+ cells accumulate (2 P, Q), suggesting that, similarly to that reported in EBV+ tumors, these cytokines are involved in PD-L1 regulation in MS brain,. ”in lines 243-246. However, it is not robust for the data from Figs. 2P and 2Q to conclude that they involved in PD-L1 regulation.
  5. The author should seriously proof-read the whole manuscript because there are lots of typos, such as “(reviewed in [45]”in line 99, “the underlying mechanisms it” in line 119, “Immunohistochemistry 20. (A)” in line 291, “Immunohistochemical 1. (A, B)” in line 312.

Author Response

Manuscript Number: ijms-3972668

EBV early lytic antigens, EBNA2 and PDL-1 in progressive multiple sclerosis brain: a coordinated contribution to viral immune evasion. By Benincasa et al.

Point-to-point replies to Reviewer #1 comments

1) RE: The English could be improved to more clearly express the research.

AU reply: The English language has been further checked by experts from our organisation who found no errors and judged the manuscript to be well written

2) RE: It is very hard to collect normal brain tissues. In order to perform a comparative research in this research, the author used lymph nodes from abdomen and axilla as well as tonsils to be control samples. They should not be suitable controls due to different tissues used between cases and controls. The author should at least explain that it is suitable to use these control samples in this research.

AU reply: We agree with the reviewer that the choice of controls is a key aspect of any research.

Our study comprises the analysis of several control tissues, both non-neurological brain specimens (all provided by the UK MS Tissue brain, see Table 2) and lymphoid tissues. Different types of control tissues were selected according to the specific experimental question being addressed. Specifically, we have used, brain tissue sections from non-neurological control cases to investigate the presence and distribution of PD-1+ and PD-L1+ cells in non-pathological cerebral tissue, and to perform the immunostaings for EBV antigens and EBER ISH to investigate the possible presence of EBV-infected cells. The results are described in our previous publication (ref.4)” Serafini et al, (2024); https://doi.org/10.1016/j.jneuroim.2024.57831”.

I addition, brain tissue sections from cases with different OIND have been also analysed as controls for the presence of EBV-infected cells in the face of a huge B cell infiltration and the results have been described in “Serafini, B et al. (2010). https://doi.org/10.1097/NEN.0b013e3181e332ec”.

Lymphoid tissue from non-infected tissue have been used as control for immunostaining of immune cells. In particular, sections from tonsils with infectious mononucleosis (IM) were used as internal control for the immunostaining of the EBV antigens. Because the immunostaining for BHRF1 was not performed before neither by us nor by others, we show the immunoreactivity for this EBV antigen in the IM tonsil (normal lymphoid tissues were negative) in the suppl. Fig. 1.

To make clearer this aspect, highlighted by Ref #1, the text has been modified in lines 151-172 and in the M&M section, lines 548-557.

3) RE: The author need to improve the presentation of data. It is confused for me to read the whole manuscript so that I don’t think I clearly understand this research.

AU reply: We are revised the manuscript in different sections to make reading the data easier. Tables and figures have been centred in the template to ensure a complete view of the results.

4) RE:  There are no title of sub-section 1 in the results section and Fig. 1 was not correctly displayed so that I couldn’t evaluate data from sub-section 1. Please correct them.

AU reply: The subsection title has now been added to the Results section, and Fig. 1 has been corrected and is now properly displayed.

5) RE: The author made a conclusion based on data “A prominent IFNγ expression and the presence of CD4+T cells producing IL-10 were detected within the meningeal TLSs where PD-L1+ cells accumulate (2 P, Q), suggesting that, similarly to that reported in EBV+ tumors, these cytokines are involved in PD-L1 regulation in MS brain,. ”in lines 243-246. However, it is not robust for the data from Figs. 2P and 2Q to conclude that they involved in PD-L1 regulation.

AU reply: The involvement of IFNγ in regulating PD-L1 expression has been described in EBV-infected cells and lymphomas in [17, 74, 81, 82]. Our aim was to investigate whether, also in cerebral tissue during progressive MS, the expression of this cytokine was present in the same inflammatory infiltrates where PD-L1+cells were more abundant. This could strengthen the hypothesis that, even in SPMS, the synergistic action between IFNγ and the inhibitory immune checkpoint PD-L1/PD-1 is established to dampen the immunocytotoxic response. Because it has been described that PD-L1 and IL-10 can work together, often in feedback loops, to repress immune responses in tumors, immunostaining for IL-10 in MS brain was performed to investigate whether cells producing this cytokine are present in the PD-L1-enriched immune infiltrates. Their presence might suggest that IL-10 contributes in establishing an immunosuppressive microenvironment locally.

Thanking the Reviewer for his/her suggestion, we have modified the text in lines 251-265, and 460-463 to improve the exposure of the objective and the results.

6) RE: The author should seriously proof-read the whole manuscript because there are lots of typos, such as “(reviewed in [45]”in line 99, “the underlying mechanisms it” in line 119, “Immunohistochemistry 20. (A)” in line 291, “Immunohistochemical 1. (A, B)” in line 312.

AU reply: We apologize for the typos that have been carefully checked and corrected in the revised version of the manuscript.

All adjustments are marked in the text.

Reviewer 2 Report

Comments and Suggestions for Authors

The authors provide an interesting view of the possible inhibitory mechanisms used by EBV during phases of the viral life cycle, supporting the hypothesis that in highly inflamed MS brain tissues containing meningeal tertiary lymphoid structures (TLSs), the virus might establish  immune-permissive habitats favouring the maintenance of an intracerebral reservoir.

The manuscript is well written and provides new evidences regarding EBV infection and involvement in MS pathophysiology. However, few stylistic arrangements might be required. 

Typos adjustments: 

  • line 119. The authors should report "and two main hypotheses have been proposed to explain the underlying mechanisms it" as "and two main hypotheses have been proposed to explain the underlying mechanisms."
  • line 166. The authors should include a brief title for the first results

Figures and tables:

  • Figure 1 should be more centered into te manuscript to better show the full image panel
  • Figure 3 should be provided in a more high quality figure
  • Figure 5 should be provided in a more high quality figure. Specifically a more visual high quality should be provided in the figure boxes providing information regarding the hypothetical mechanisms involved in EBV persistence in SPMS brain
  • Table 2 should be centered into the manuscript to better show all information provided into table columns.  

Author Response

We thank the Reviewer for the appreciation expressed for our work.

Typos adjustments required: 

1) RE: line 119. The authors should report "and two main hypotheses have been proposed to explain the underlying mechanisms it" as "and two main hypotheses have been proposed to explain the underlying mechanisms." line 166. The authors should include a brief title for the first results

AU reply: We thank the Reviewer for these observations. Typos adjustments have been done as requested.

2) RE: Figures and tables: Figure 1 should be more centered into the manuscript to better show the full image panel

AU reply: Figure 1 has been centred in the template.

3) RE: Figure 3 should be provided in a more high quality figure

 AU reply: A high quality figure 3 has been included in the revised version. 

4) RE: Figure 5 should be provided in a more high quality figure. Specifically a more visual high quality should be provided in the figure boxes providing information regarding the hypothetical mechanisms involved in EBV persistence in SPMS brain

AU reply: A high quality figure 5 has been included in the revised version, according to the suggestion. 

5) RE: Table 2 should be centered into the manuscript to better show all information provided into table columns.  

 AU reply: Table 2 has been centred in the template.

For the reviewers

Other changes have been made in the manuscript to improve it, as requested by both Reviewers:

  • Figure 1: legend has been improved
  • Figure 3 and its legend have been replaced with the corrected versions
  • One panel has been added to figure 4 (inset in A) and legend has been updated
  • Tab 1 has been formatted to fit better into the template.
  • Ref 17 and 53 have been completed

All adjustments are marked in the text.

Round 2

Reviewer 1 Report

Comments and Suggestions for Authors

No more concern.